# Functional Post-Synthetic Chemistry of Metal–Organic Cages According to Molecular Architecture and Specific Geometry of Origin

**DOI:** 10.3390/molecules30030462

**Published:** 2025-01-21

**Authors:** Rodrigo Cué-Sampedro, José Antonio Sánchez-Fernández

**Affiliations:** 1Escuela de Ingeniería y Ciencias, Tecnologico de Monterrey, Ave. Eugenio Garza Sada 2501, Monterrey 64849, Mexico; 2Procesos de Polimerización, Centro de Investigación en Química Aplicada, Blvd. Enrique Reyna No. 140, Saltillo 25294, Mexico

**Keywords:** metal–organic cages, supramolecular chemistry, polyhedra, metal–organic polyhedra, synthesis of metal–organic cages, self-assembled molecules, porous coordination cages

## Abstract

Metal–organic cages (MOCs) are discrete supramolecular entities consisting of metal nodes and organic connectors or linkers; MOCs are noted for their high porosity and processability. Chemically, they can be post-synthetically modified (PSM) and new functional groups can be introduced, presenting attractive qualities, and it is expected that their new properties will differ from those of the original compound. This is why they are highly regarded in the fields of biology and chemistry. The present review deals with the current PSM strategies used for MOCs, including covalent, coordination, and noncovalent methods and their structural benefits. The main emphasis of this review is to show to what extent and under what circumstances a MOC can be designed to obtain a tailored geometric architecture. Although sometimes unclear when examining supramolecular systems, particularizing the design of and systematic approaches to the development and characterization of families of MOCs provides new insights into structure–function relationships, which will guide future developments.

## 1. Introduction

Historically, the existence of a discrete relationship between metal–organic frameworks (MOFs) and supramolecular architecture based on metal–organic polyhedrons (MOPs) has been reported. In fact, microporosity is an important asset in supramolecular hybrid metal–organic cages, also called porous coordination cages (PCCs). In this sense, the characteristic crystalline phase of a MOP depends on the nature of the cage as much as on the conditions of crystallization [1]. It is relevant that in pore technology, peptide-based nanopores are also exploited, with potential application in molecular detection [2].

Rapid advances associated with the design and synthesis of polyhedra using molecules and metals through coordination-driven self-assembly have been significant, including the introduction of new strategies of synthesis. When polyhedral structures are designed employing inorganic and organic building blocks, the vertices are either polytopic inorganic or organic nodes, and the edges are the linkers between the inorganic nodes and between inorganic and organic nodes. Spatially, polyhedral structures must be built from at least one block forming a bend or curve, and this obviously implies the good design of a supramolecular polymerization reaction [3].

MOPs have been defined as discrete molecules self-assembled from organic linkers and metal clusters, and they are recognized for having tiny porous units. From an instructive viewpoint, metal–organic materials can be categorized into four distinct configurations: uniform structures, heterostructures, hierarchical structures, and structures with gradient.

Giant metal-containing molecules were considered by Virovets et al. [4], and considering such studies there are two relevant classes: a giant cluster with a compact core surrounded by ligands, and spacer-based supramolecules that comprise metal cations or polynuclear metal complexes (nodes) connected to each other by organic ligands as the spacers. A giant hollow structure, for instance [Pd_30_(L^1^)_60_]^60+^, is often referred to as a coordination cage, metal–organic cage (MOC), or metal–organic polyhedron (MOP) [4].

Growing interest was made apparent in a study 2016 [5], which precisely highlighted that the metal atoms in the central core can have different cohesive energies depending on the elements, the nature of the outer ligands, and the surface energy. The overall conclusion was that the interrelations of different kernel chemical transformations of one kernel into another, as well as the interplay between a kernel and the set of frequently occurring numbers of the metal atoms in the kernel (so-called magic numbers or magic series 12,115,116) [5], are important aspects of the chemistry of giant metal clusters. For instance, for the Au/TBBT (TBBT = 4-tertbutylbenzenethiolate) arrangement, the magic series includes clusters of the general formula Au_8n+4_(TBBT)_4n+8_, where n = 3–6.

One of the pioneering works on polyhedral structures presents three possibilities in a process called truncation for square units, namely, a truncated octahedron with 6 squares, a truncated cuboctahedron with 12 squares, and a truncated icosidodecahedron with 30 squares [6].

It is now reasonably certain that the careful selection of ligands and metal centers gives rise to varied geometrical convex or nonconvex polyhedral shapes of solids. Interesting comments from El-Sayed and Yuan describe convex polyhedra, and they detail the formation of polyhedral MOP with specific and detailed geometric architecture, such as regular polyhedra that form Platonic MOP and semi-regular polyhedrons that favor the formation of Archimedean MOP; these two types of polyhedra are formed by edge-sharing of their polygons. One significant example is the formation of supramolecular coordination structures through metal-mediated self-assembly and calixarene units. Simply combining linear di- or tricarboxylate or azole ligands with metal–calixarene units gives rise to multicomponent hollow MOC structures with adaptable shapes and distinctive features [7]. A particularly influential paper by Su and coworkers [8] detailed the symmetrical elements, inversion center, and plane existing in polyhedra, properties that have been quite a challenge in obtaining chiral MOCs with various configurations. The stereogenic centers in polyhedral MOCs can be included at the vertices, edges, or faces of the cage entities before the coordination self-assembly process, or they can alternatively be formed in situ through the spatial arrangement or twisting configuration of either metal centers or organic ligands.

There are only five kinds of Platonic solids, namely tetrahedron, cube (hexahedron), octahedron, dodecahedron, and icosahedron. A regular polyhedron has a high symmetry group, comprising multiple symmetrical axes and mirror planes (and inversion center). In addition, there are all together thirteen types of Archimedean solids, most with *Td*, *Oh*, or *Ih* symmetry groups, except that snub cubes and snub dodecahedrons are in the O and I symmetry groups, respectively, as shown in Figure 1.

The value of a library of structurally symmetric compounds in the study of the effect of geometry, size, and periphery is self-evident. This approach is also taken in the mathematical concept of face-rotating polyhedra (FRP), which heralds a fascinating category of chiral objects. Further, in FRP, each face rotates around its central axis, giving rise to distinct facial rotational patterns while maintaining the core polyhedral shape [9], as shown in Figure 2. A systematic variation in the symmetry becomes possible by a design process, which, in a formal sense, includes octahedral geometry with different stereoisomeric rotation patterns.

A major obstacle to designing FRP is identifying the molecular blocks that may have specific rotation controls. Considering the mathematical concept in molecular architecture will be the foundation for a significant class of chiral molecular polyhedra. The rapid progress made over the time in developing novel MOP, this distinction is revealed by the assembly of the compound [Rh_2_(bdc-C_12_)_2_]_12_ (C_12_RhMOP; bdc-C_12_ = 5-dodecoxybenzene-1,3-dicarboxylate) with the ditopic linker, 1,4-bis(imidazole-1-ylmethyl)benzene (bix) [10].

An alternative to the permanent porosity of polyhedral organometallic structures has been published by Yagui and his team of collaborators [11]. They reported the schematic synthesis of both MOP-100 and MOP-101 (so named by the authors) structured with Pd^II^, resulting in rhombic dodecahedra, and moreover established that the compounds presented high chemical stability in acid and basic environments. The MOP-100 was assembled with hydrogen tetrakis(1-imidazolyl)borate (HBIm_4_) and the MOP-101 was assembled with hydrogen tetrakis(4-methyl-1-imidazolyl)borate (HB(4-mIm)_4_). One particular is that in which a pre-assembled complex of the cuboctahedral MOP is used as a possible template to replicate a caged structure with both a heterometallic and heteroleptic “triblock Janus-type” configuration. This easily visualized by considering a strategy to form the archetypical cuboctahedral Cu_24_bdc_24_ or Rh_24_bdc_24_ MOP (here, bdc is the compound 1,3-benzenedicarboxylic acid), where the triblock Janus-type configuration is conceived [12].

To generate heteroleptic coordination cages with a potential application in guest recognition, chemical sensing, and catalysis, chirality is a significant factor. Structural assessment by NMR methods and single-crystal X-ray diffraction is essential, and it is necessary to consider that during the synthesis, a metal precursor and the ligand are dissolved and heated until the desired cages have assembled, favoring the formation of more thermodynamically stable products [13].

Notably, a wealth of applications have been reported by Cook and Morrow and their colleagues on the variations in the molecular geometry of the geometrical criteria of a MOP chemically constituted with rigid catecholamide linkers for a M_4_L_6_ MOP utilizing multiple Fe^III^ centers [14]. This all relies on the idea that the essential factor is stabilization and its manifestation. The intrinsic nature of self-assembly is kinetically inert in the presence of Zn^II^ cations and EDTA for up to 24 h, dissimilar to simple catechol complexes of Fe^III^ that rapidly decompose under such conditions. An additional advantage is that structures are available to bind strongly to serum albumin, which has an impact on the in vivo pharmacokinetic properties observed in mouse models and in the increases observed with T_1_ relaxivity.

## 2. Structural Criteria of MOP

If we think of the structural criteria for coordination-driven self-assembly as a powerful method to create MOP from supramolecular complexes, the design needs a combination of both “donor” organic bridging ligands and suitable “acceptor” metal ions or discrete metal–oxo clusters in the corners to yield a variety of architectures. Encouraged by the success of all the possible configurational isomers of metal–organic polyhedron-based structures using the Bi_6_Fe_13_L_12_ cluster as a model, Kandasamy et al. [15] showed that ligand variation plays a leading role during the assembly process in the design of a system based on Bi_6_Fe_13_L_12_. In a way, the supramolecular aggregation of cationic {Bi_6_Fe_13_L_12_}-type clusters favors the impact on the pKa.

One of the most fundamental and earliest criteria of MOFs and their active sites are the structural rigidity and inter-channel connectivity, which are not present in MOPs that present random aggregation or reorganization during solvent removal due to a strong supramolecular interaction between neighboring cages restricting the diffusion of sorbates through the MOP [16]. The bonding existing in MOF- and MOP-based supramolecular frameworks is remarkably maintained when diverse porosities are anchored. Such pore diversity can be observed in the tetrahedral pores of UiO-66, replicable in a discrete MOP. Specifically, an extended supramolecular framework of the UiO-66 type contains pores similar to those present in amine-functionalized ZrMOP [17].

We must be consistently cautious and circumspect, always carefully and deliberately avoiding describing the molecules as straightforward synthetic strategies in the hydrolytic stability of MOPs made up of carboxylates, invariably preferring instead to refer to them as supramolecular building blocks (SBBs), an efficient route to restricting any aggregation of a MOP in the solid state after guest removal. In fact, as Ghosh and his colleagues explicitly pointed out, MOPs are separated from each other by strong coordination bonds that resist any reorientation of the MOPs upon guest removal [16].

### 2.1. Geometric Value of MOP

A related rigorous approach was taken by Lai et al., who sought a method based on the logarithmic kinetics of surface ligand exchange on an MOP surface. This is extremely important information since it is connected to polymer–polymer interaction observable within a plausible time scale. To this point, the authors specify the preparation of functionalized 24-arm MOP-based miktoarm star polymers (MSPs, also known as *μ*-star polymers), where temperature is a decisive factor [18]. The behavior towards a new vision founded by Kondinski et al. [19] involves the development of new concepts of not only chemical but also a geometric nature. This is through the representation of a chemical building unit (CBU) and another geometric building unit (GBU), the latter of which is the basis of the construction of MOP assembly models formulated through complementary CBUs linked to an initial GBU, as can be seen in Figure 3.

In the meantime, a colossal amount of evidence has created a very strong cause and effect link between π-delocalization and geometry, with a lot of emphasis on the preference of the π-electron energy. This was categorized in a study of three MOPs and their respective specific geometries, such as a Zr-based MOP with four [Cp_3_Zr_3_(*μ*_3_-O)(*μ*_2_-OH)_3_]^4+^ metal nodes formed by the hydrolysis of zirconocene dichloride and six 1,4-benzenedicarboxylate linkers, which bring about a tetrahedral geometry; a MOP based on Cu formed with twelve Cu–Cu paddlewheel nodes at the vertices bridged by 1,3-benzenedicarboxylate linkers, which results in a truncated–cuboctahedral geometry; finally, the geometry of a MOP formed by six Pd^2+^ nodes at the vertices linked by four 2,4,6-tris(4-pyridyl)-1,3,5-triazine (TPT) ligands on the faces presents a truncated tetrahedral structure, and it is worth mentioning that the sites of coordination of the Pd metal centers in this MOP are capped by 2,2′-bipyridine groups to give a discrete structure instead of an extended framework [17]. Following the structural bases, the studies conducted by Kwon and Choe follow the structuring of MOP-1-R ([(Cp_3_Zr_3_O(OH)_3_)_4_(BDC-R)_6_]Cl_6_[(CH_3_)_2_NH_2_]_2_, Cp = h^5^-C_5_H_5_, H_2_BDC = benzene-1,4-dicarboxylic acid, R = CH_3_, OH, and Br), with the construction of the forming blocks comprising four Zr clusters and six BDC-R linkers at the vertices and edges of the tetrahedral cages [20].

In the same perspective, Ma and his team of collaborators [21] present the synthesis of a bifunctional discrete metal–organic cuboctahedron, namely Cu-MOP, from copper acetate and 2,6-dimethylpyridine-3,5-dicarboxylic acid, with Lewis acidic and basic functional centers. Cu-MOP has effective catalytic properties in Knoevenagel deacetylation and Henry deacetylation reactions.

### 2.2. Importance of Porosity

The porosity of compacted molecules in the solid state was particularly explained by Hasell and Cooper [22]. They detail the potential for directional intermolecular interactions such as hydrogen bonding, which can generate low-density and energetically stable chemical architectures, for instance, porous cages such as MOFs and MOCs.

A material designed with the property of retaining porosity after melting and cooling cycles assessed through CO_2_ adsorption and desorption experiments at 298 K was specified by Baeckmann et al. [23], who also reported that such a porous material, i.e., an Rh^II^-based octahedral form, was functionalized with polyethylene glycol. Such a material, once melted, is very likely to be a vehicle for dispersing other materials and even molecules in order to design mixed matrix composites.

A less often used criterion is based on the thermodynamic behavior of structures that have minimal or no empty spaces, emerging as metastable products if the reaction is under strictly kinetic considerations. This is particularly true even if we also take into consideration structural and functional compatibility among building blocks that are positioned appropriately to each other [21]. As Sullivan et al. [24] suggested, in the isolate pure phase of the material [ZrMOP-biphenyl]OTf_X_ in particular, the tetrahedral structure [tetZrMOP-biphenyl]OTf_4_ was isolated in just 20 min, while the so-called lantern structure (also called a cigar-type architecture) was isolated in a reaction time greater than 30 min; in addition, the solvent factor is undoubtedly a decisive factor in self-assembly.

## 3. MOC Synthesis Criteria

There is particular interest in distinguishing an efficient strategy for the research of non-classical polyhedra cages with captivating architectures for molecular recognition and separation, sensing, and supramolecular catalysis, and this is the inspiration for advanced tailoring forms [25]. This assumption poses certain convergence with the symmetry of macromolecular structures, and their properties can help deliver key information in several modern fields such as those related to biomolecules and molecular electronics, giving valuable insights into high-yield coordination chemistry-based applications that allow us to tailor the size, shape, and properties of the resulting architectures [26]. An important remark in this area was made by Hosono and Kitagawa, who recognized that the structural relationship between the highly symmetrical shape of MOPs and the geometrical positioning of the metal ions and ligands makes them suitable building blocks to fabricate MOP-core macromolecules as discrete porous modules [27]. MOPs are formed by the self-assembly of multiple metal ions and organic ligands, as shown in Figure 4a. The underlying challenges in exploiting MOPs are their low solubility, low processability, and high crystallinity due to intermolecular interactions, but these can be overcome by coupling of MOPs with polymers to improve their solubility and endow them with a thermoplastic property to make them processable, as exemplified in Figure 4b [27].

The skeletal bonding topology in electron-rich polyhedra give rise to the porous coordination cages also known as MOPs, with the intrinsic pores being a subclass of supramolecular cages that can be constructed from metal cations and organic linkers by using a modular approach [28].

A comprehensive method to generate a porous MOC liquid by incorporating PEGimidazolium chains into the periphery of an MOC was specified by He et al. [29]. As coulombic repulsion can keep chains positively charged, the cavities were terminated by the positive imidazolium moieties. PEG–imidazolium chains were integrated into a p-tert-butylsulfonylcalix [4]arene-based MOC to form a porous liquid Im-PL-Cage, which was self-assembled by the ionic ligand PEG-imidazolium 1,3-benzenedicarboxylic acid (PEG-Im-H_2_BDC), Zn(NO_3_)_2_·6H_2_O, and a p-tert-butyl-sulfonylcalix [4]arene (H_4_TBSC) via coordination bonds. The long PEG chains can not only shield, guaranteeing the accessibility of the host cavities, but can also lower the melting point of the Im-PL-Cage and cause it to behave as a liquid [29].

The self-assembly of a spherical structure was composed of 30 palladium ions and 60 bent ligands consists of a combination of 8 triangles and 24 squares and had a symmetrical pattern similar to tetravalent Goldberg polyhedron. We must not forget that Platonic and Archimedean solids have been prepared through self-assembly, as have trivalent Goldberg polyhedra, which occur naturally in the form of virus capsids and fullerenes. Using graph theory to predict the self-assembly of even larger tetravalent Goldberg polyhedra more stably will enable the polyhedron family to be assembled from 144 components, namely, 48 palladium ions and 96 bent ligands [30]. Platonic solids such as the Archimedean solids are characterized by having equivalent types of vertices surrounded by regular polygons.

The octahedral (M_2_)_6_L_12_-based MOP structure was first constructed from a Cu_2_ paddlewheel and 2,2′:5′,2″-terthiophene-5,5″-dicarboxylate, and it was explored by bridging various metal paddlewheel clusters with a 9*H*-carbazole-3,6-dicarboxylate linker [31]. Another observation by Hirscher and Cooper and collaborators was the protection–deprotection strategy to produce cages where five out of the six internal reaction sites in the cage cavity were functionalized by means of formaldehyde. The combination of small-pore and large-pore cages together in a single solid produces an optimal material for the separation of deuterium and hydrogen with a selectivity of 8.0 and with a high deuterium uptake of 4.7 millimoles per gram [32].

A remaining challenge is to advance in the synthesis of discrete molecular cages with distorted structures by the self-assembly of asymmetric building units. The potential of each chemical factor and physical parameter is key in the formation of cages, and the lability of the metal ion used is decisive; in conclusion, the reaction temperature is relevant. For instance, cages formed with Pt^II^ require more temperature than cages obtained from Pd^II^, or cages formed with Rh^II^ require more temperature than cages obtained from Cu^II^. Sometimes, as appropriate, coordinative solvents are used to favor the reversibility of the formed coordination bonds [33].

In an example illustrated by Banerjee et al., a class of compounds involving functionalized benzothiadiazole units of twisted di-terpyridine ligand L with 4-pyridyl donor sites was structurally dimensioned. With possible self-assembly using a Pd^II^ acceptor (A), L assumes a trans orientation of its terpyridine units to generate a distorted trigonal Pd_6_ cage. Ligand L was synthesized following Krönhke pyridine synthesis by the reaction of 3,3′-(benzo[c][1,2,5]-thiadiazole-4,7-diyl)dibenzaldehyde (P) with KOH, 4-acetylpyridine and NH_3_(aq) in ethanol. The final cage was synthesized via the self-assembly reaction between L and cis-[(tmeda)Pd(ONO_2_)_2_] (A) in 1:2 molar ratio in DMSO at 70 °C for 48 h (tmeda = *N*,*N*,*N*’,*N*’-tetramethylethane-1,2-diamine; DMSO = dimethyl sulfoxide) [34].

There are other cases in the organic synthesis of some MOCs that use o-phenanthroline and its derivatives to generate functional organic ligands to form metal–organic materials with favorable properties in fluorescence detection. Since o-phenanthroline has coordination properties with transition metals, the compound 3,3′-[(1E,1′E)-(1,10-phenanthroline-2,9-diyl)bis(ethene-2,1-diyl)]-dibenzoic acid and cadmium salts were structured, exhibiting a trefoil-shaped form. The structural congruence obtained was evaluated by single-crystal X-ray crystallography and intramolecular hydrogen bonding studies [35]. Craig et al. [36] described an alternative containing a bicyclo [2.2.2]oct-7-ene-2,3,5,6-tetracarboxydiimide unit for assembling lantern-type metal–organic cages with the general formula [Cu_4_L_4_].

Research has been carried out that even takes advantage of the expansion of some MOCs designed with L dicarboxylates with a rigid aromatic skeleton, in which the two benzene rings of the stilbene moiety are connected by a hexaethylene glycol chain at the 2,2′ positions. The relevance of these materials is expanded by using Rh^II^ to structurally dimension a conformation of a paddlewheel cage with great stability and versatility [37].

In another instance, from an investigation directed by Furukawa [38], analysis of the reaction mechanism of a MOP with imidazole-based linkers disclosed the polymerization to consist of three separate stages, namely, nucleation, elongation, and cross-linking. The authors use supramolecular polymerization to drive the monomeric succession of MOP monomers. A summary of the history, dynamics, and complexity of the conformation of MnL_2n_ cages is described by Judge et al. In the meantime, progress is also substantial in exploiting self-assembled MOCs as drug carriers. In addition, a central point is given to the methods of integrating MOCs in polymers, leading to the synthesis of colloidal nanoparticle micelles and vesicles that possess combined properties and extended system tunability [39].

## 4. Structural Post-Modification of MOCs

In a comprehensive work published in 2021 by Martín Díaz and Lewis, the complexity of MOCs over thirty years is quoted [40]. With the introduction of new ideas in this regard [13], both heteroleptic assemblies and low-symmetry [41] assemblies towards the development of more sophisticated host systems are more commonly faltering. The combination of the above together with the great improvement in single-crystal X-ray diffraction (SCXRD) and the advancement of computational power for theoretical investigations allows researchers to gain an in-depth analysis of these systems.

Optimized geometries’ supramolecular binding to guests via an endohedral functionalization is an attractive alternative in employing a covalent linker as the endohedral group. A representative case is the linker dependence of the electron transfer of redox-active species encapsulated in M_6_L_12_ and M_12_L_24_ supramolecular cages [42].

### Relationship Between MOFs and MOCs Linked to Post-Synthetic Modification Strategies

A key factor in the modification of a series of calixarene-protected structures was exposed because they can form tunable and stable vertices necessary for post-synthetic modification (PSM) approaches [43]. This conveniently allowed the group of Chen and Zhang to synthesize advances in the considerations raised in PSM reactions exclusively in pre-assembled MOCs, leaving aside the structural dissociation and reintegration of PSM processes (which as such as are building block exchange ligand/metal reactions); structural transformations instead of post-assembly triggered by various stimuli; and host–guest chemistry inside of the MOCs cavities [44]. On the contrary, they emphasize covalent and coordination modification in terms of PSM structural variation in both the exterior and interior; nevertheless, noncovalent strategies center only on exterior PSMs. Another important characteristic that the authors discuss is post-modified products; in this regard, they make a relevant division of 0D cages to 0D cages; 0D cages to one-dimensional (1D) or two-dimensional (2D) structures; 0D cages to three-dimensional (3D) network materials [44], as exemplified in Figure 5. It has even been suggested that the use of pristine, post-synthetically modified MOFs to generate catalysts for polycarbonate (PC) depolymerization by methanolysis reaction is possible [45].

Comparing MOFs to MOCs that possess well-defined structures in a well-defined coordination environment, the 0D structure of MOCs allows for the exposure of more active sites during catalytic reactions [46].

The PSM is well developed in the fields of metal–organic frameworks (MOFs), covalent organic frameworks (COFs), and porous organic cages [47]. In the context of supramolecular coordination chemistry, Nitschke et al. [48] have suggested a post-assembly modification (PAM) process comparable to the PSM concept. This makes possible the construction of a metal–supramolecular complex requiring metal coordination self-assembly equilibria, leading to the construction of higher-order molecular architectural blocks. In the attempt to construct a truncated tetrahedral cage, a dendritic organic–metal ligand was constructed by coordinating tripodal organic ligands with Ru^2+^. Subsequently, the Ru building blocks precisely generate the desired organic–heterometallic cages by complexation with Fe^2+^ [49]. On the contrary, we must not forget that MOCs are sensitive to solvents, especially those possessing coordination capabilities that significantly alter the structural arrangement of the cage [50]. Strategies to elaborate MOCs have drifted over time, and their structural modification is influenced by synthesis and its versatility (Figure 6). It is of consideration that the presence of various reversible interactions in MOC systems may be easily disrupted during the PSM process. This characteristic considers the orthogonality between reactions and the supramolecular interactions in the MOCs [51]. A post-synthetic covalent modification strategy favored by a [4+2] cycloaddition between anthracene and maleimide under mild conditions to obtain MOCs with tailored functions is well observed because they are true platforms to implant others functional groups, namely, cyclohexyl, α-phenethyl, a chiral binaphthol moiety, tetraphenylethylene group, and pyrene substituent through the covalent PSM process [51].

Coordination bonds within coordination chemistry are attractive largely because of their potential in generating porous materials for the confinement of varied molecular species with engineered molecular affinity. The reason for this behavior in the case of the coordination cages (CCs) or MOCs is orchestrated by the smallest units to form large molecular networks. The fine balance between this coordination and the molecular geometry was described in 1990 by Fujita et al. [52] and Tateishi et al. [53] recently.

The controlled post-synthetic functionalization of MOP using coordination chemistry on metal ions and covalent bonds in organic linkers was exploited in Rh^II^-based materials with cuboctahedral geometry for the high microporosity of the HRhMOP of 947 m^2^/g and that of Rh-OH-MOP of 548 m^2^/g, i.e., [Rh_2_(bdc)_2_(H_2_O)_2_]_12_ and [Rh_2_(OH-bdc)_2_(H_2_O)_1_(DMA)_1_]_12_, where OH-bdc = 5-hydroxy-1,3-benzenedicarboxylate and DMA = *N*,*N*-dimethylacetamide), as they can withstand aggressive reaction conditions, including high temperatures and the presence of strong bases [54].

The need to consider intermolecular conjugational effects is further highlighted by the fact that a post-modification protocol requires covalent conjugation of a chiral cholesteryl pendant to a MOP. Both experimental and computational measurement results have validated the role of intercholesteryl forces in MOPs, which achieved chirality transfer to a supramolecular scale with chiral optics. It should be mentioned that the formation of a supramolecular chirality is associated with cholesteryl groups as well as the induced helical packing. The individual alternative self-assembly of MOPs produces achiral crystalline plates due to the absence of stereocenters [55].

It is also noteworthy that in applications of the multi-fold post-modification of macrocycles and cages, introducing functional groups into two- and three-dimensional supramolecular scaffolds bearing fluorinated substituents [56] is associated with the activation of strong C-F bonds [57], opening possibilities in multi-step supramolecular chemistry to employ readily available isocyanates. In a particular case, benzylamine was reacted in an isothiocyanate-promoted cyclisation with azadefluorination. The difluoro substitution led to the formation of a cyclic six-membered urea (1,3-diazinan-2-one), enabling exo-functionalization of the supramolecular entities [58].

In another study, an alkyne-based hydrocarbon cage was synthesized using alkyne–alkyne coupling in the cage-forming step. The reaction of the cage with Co_2_(CO)_8_ resulted in metalation of its 12 alkyne groups to give the Co_24_(CO)_72_ adduct of the cage [59]. In this same consideration and in support of the complementary techniques of ^1^H-NMR, electrospray ionization mass spectrometry (ESI-MS) and 1D-exchange spectroscopy (EXSY) were used in the investigation of pyridyl ligand substitution of the model complex M(py)_2_ (M = (*N*,*N*,*N*′,*N*′-tetramethylethylenediamine)palladium^II^, py = pyridine). Basically, using different temperatures, ^1^H-NMR, and 1D-exchange spectroscopy (EXSY), the reaction rates and activation energies (*E*a) for pyridyl ligand substitution under different reagent and reactant conditions were determined [60].

Holistic considerations such as environmental considerations are an essential part of the efficient PSM of MOCs to fabricate catalysts that are not accessible under conventional synthesis conditions, for instance, stabilizing highly active but vulnerable catalysts to achieve high-efficiency catalysis.

In a communication, Pausch et al. specified the multiple post-modifications of macrocycles and cages, introducing functional groups into two- and three-dimensional supramolecular structures with fluorinated substituents, opening new possibilities within supramolecular chemistry. The chemical essence of this structural modification utilizes benzylamine- and 1,4-diisocyanatobenzene-promoted cyclization [58].

Two homochiral porous organic cages of imine condensations of tetrapic 5,10-di(3,5-diformylphenyl)-5,10-dihydrophenazine and ditopic 1,2-cyclohexanediamine yielded two chiral [4+8] organic cage isomers with absolutely different topologies and geometries depending on the orientations of the four tetraaldehyde units relative to each other. One isomer had a Johnson-type structure while the other embraced a tetragonal prismatic structure [61].

It is appreciated that scientific reports on enantioselective structural isomerization are important for the substantial advances in the design and construction of diversified functional supramolecular systems.

## 5. Specificity of MOCs

The ability of ligands and metal ions/nodes to form complex self-linked molecular networks is as broad as that of any supramolecular system; discrete, porous architectures are remarkable given the simplicity of their components and the relative ease of their synthesis. In this context, pioneering works have taken advantage of the high porosity of Pd_12_L_24_ nanospheres, allowing the entry and exit of a variety of reagents useful for general chemical transformations [62]. Such combinations can be further sophisticated to form structures with an evident selectivity towards designer chemical transformations. The next important contribution in this area divulges the synthesis of a Pd_2_L2_4_-type MOC (L2 = 5-Azido-*N*,*N*’-di-pyridin-3-yl-isophthalamide) and its ability to produce the supramolecular metallogel (PdG); this compound was subsequently loaded with DOX and successfully delivered to B16–F10 melanoma cells. Furthermore, the metallogel was characterized by dynamic rheology, showing a reversible property conducive to topical application [63].

Similar ideas to those applied by Bera et al. [63] in relation to metal–organic cages could be used as cross-linkers in microgels or nanogels. This implication is demonstrable in palladium organic cages with acrylamide side chains produced by precipitation polymerization. The resulting nanogels may favor applications such as specific sorbents for chloride ions and for the reactive release of the anticancer drug abiraterone [64].

Another criterion specified by Berber et al. [65] used a seven-residue proline design (trans-4-hydroxyproline)-(proline)-5-(trans-4-hydroxyproline) using a solid-state microwave-assisted technique, with a tert-butyl carbonyl group, to avoid competitive binding of a free amine to palladium(II) and to act as a reporter signal in ^1^H nuclear magnetic resonance (NMR). That decision was made in consequence of previous reports suggesting that six residues were the minimum required to ensure the stable formation of a polyproline (PPII) helix in aqueous solution, with a repeat length of approximately 9 Å, where every third residue aligns. Avery, Algar, and Preston [66] detailed the different methodologies and complexities of lantern cages. For example, they described the different ligands used to form them, such as low-symmetry ligands (homoleptic arrays), different metal ions (heterometallic arrays), and differentiated cavities (multicavity arrays). This may anchor the symmetric geometries into self-assembled metallo-supramolecular systems of lantern cages.

### 5.1. The Significance of Hydrogels in the Structuring of MOCs

Recently, Roy et al. [67] opened possibilities to the distribution of stress relaxation time scales in metal-coordinated hydrogels with a variety of cross-linker sizes, including ions, MOCs, and nanoparticles. The same characteristic patterns on the guests selected in differential spaces within polymeric hydrogels cross-linked with metal–organic cages was one of the contributions my by Küng et al. [68] when designing the synthesis of a series of gels based on metal–organic cages with varied bond lengths and ranging from an average molar mass of 1 to 6 kDa; for graphical detail, see Figure 7.

Stepwise deformation experiments of such polyethylene glycol (PEG)-based gels with a variety of cross-linking valencies were performed, and their corresponding relaxation curves were evaluated. Roy and co-workers [67] have prompted nitrocatechol-functionalized PEG whose cross-linking is cemented by Fe^3+^ ions and by iron oxide nanoparticles with an average diameter of 7 nm and a surface area allowing for a valency of ~100 ligands. Another set of gels was made with bispyridine-functionalized PEG, where bis-meta-pyridine ligands induced the self-assembly of gels that were cross-linked by Pd_2_L_4_ nanocages; in another instance, bis-para-pyridine ligands induce the self-assembly of gels that are cross-linked by Pd_12_L_24_ nanocages. On the other hand, Sutar et al. [69] establish that a hybrid material incorporating self-assembled metal–organic cages in a gel matrix could result in an excellent proton conductor. These authors elucidate that in the open supramolecular framework {(Me_2_NH_2_)_12_[Ga_8_(ImDC)_12_] DMF∙29H_2_O} (ImDC = 4,5-imidazole dicarboxylate), MOCs with a [Ga_8_(ImDC)_12_]^12−^ structure are linked together by a dimethylammonium (DMA)-assisted hydrogen bonding interaction, where free carboxylate groups on the surface of the MOCs (~24 per MOC) together with extensive hydrogen bonds between the host water molecules and DMA cations around the MOC contribute to an enviable conductivity of (2.3 × 10^−5^ S cm^−1^) under ambient conditions.

As part of their study designed to elucidate a class of gels assembled from polymeric ligands and metal–organic cages (MOCs) as junctions, Johnson and co-workers presented evidence for the formation of gels that can be finely tuned and may exhibit increased branching functionality. Namely, they presented a low-network-branch functionality polyMOC based on an M_2_L_4_ paddlewheel cluster junction and a higher-network-branch functionality compositionally isomeric one based on an M_12_L_24_ cage [70].

### 5.2. Photoswitching Structures

In another order of ideas, Calvo-Lozano et al. argue that there are several examples that take advantage of the molecular recognition of discrete molecular cages to design optical sensors, in particular to create a Rh-MOP-functionalized bimodal waveguide (BiMW) sensor for the real-time detection of water contaminants such as 1,2,3-benzotriazole (BTA) and the insecticide imidacloprid (IMD) in less than 15 min, presenting a limit of detection (LOD) as low as 0.068 μg/mL for BTA and 0.107 μg/mL for IMD [71]. Also, advances in modulating the conformation of stimuli-responsive MOCs have been structured with bis-pyridyl dithienylethylene (DTE) groups. The presence of Pd^2+^ is important to assist the reversible photoswitching between small Pd_3_L_6_ rings irradiated with green light and large Pd_24_L_48_ rhombicuboctahedra when exposed to ultraviolet (UV) light [72]. Figure 8 indicates a route to the design of materials with switchable topology (Figure 8a) and to a synthetic design of a poly(ethylene glycol)-based polymeric ligand (Figure 8b).

UV-induced changes in arylazopyrazoles in the presence of a self-assembled cage based on Pd-imidazole coordination have been the subject of systematic study. In this context, the isomerization of the *E*-isomer of arylazopyrazole, which is not water-soluble by itself, has been exploited in aqueous media. NMR spectroscopy and X-ray crystallography have demonstrated, in a complementary manner, that each cage can encapsulate two *E*-arylazopyrazole molecules. The UV-induced switch to the *Z*-isomer was accompanied by the release of one of the two guests from the cage and the formation of a 1:1 cage/*Z*-arylazopyrazole inclusion complex. DFT calculations unambiguously suggest that this process involves changes in the cage conformation. In their calculations, Hanopolskyi et al. [73] argue that a retro-isomerization is induced with green light, resulting in the initial 1:2 cage/*E*-arylazopyrazole complex.

Critical self-assembly with Pd^II^ cations leads to the formation of Pd_n_L_2n_ supramolecular architectures. With the uniqueness of using diazocine in stable cis isomeric formation, structural strain can be an obstacle in the formation of a defined structure, i.e., a unique structural product. Clever and co-workers [74] pointed out that diazocine in the metastable trans isomeric form assembles into a lantern-shaped cage design [Pd_2_(*trans*-L)_4_] as an exclusive species.

In further technological advances, Hosoya et al. [75] proposed absolute hardness as they synthesized a host-responsive, luminescent MOC composed of porphyrin dyes and Yb^III^ complexes. The MOC possessed the defined three-dimensional cavity to intercalate small planar aromatic perfluorocarbons inside the cage, creating 1:1 host–guest supramolecules, as detailed in Figure 9a,b. The authors argued that the host–guest packing partially inhibited oxygen molecules from approaching the cage, thereby enhancing the near-infrared (NIR) fluorescence of Yb^III^. It is quite remarkable that a diazocine moiety was implemented in the main chain of a banana-shaped bis-pyridyl ligand in order to fully sustain a photoswitching specificity.

The properties of the open and closed forms’ structures are evidently as special as the presence of chiral guests influencing the enantioselectivity and the structural opening and closing. Not only do the open and closed forms have different guest-binding properties, but the presence of chiral guests can also influence the enantioselectivity of the ring’s open and closure switching [76]. There have been developments in light-promoted photoswitchable MOC structures, whose architecture is organized with dithienylethylene units undergoing reversible forms of interconversion, i.e., closed and open forms, as illustratively described in Figure 10a,b.

The convenience of using diverse stimuli-responsive molecules such as spiropyran, spirooxazine, chromene, azobenzene, and diarylethene derivatives in combination with a variety of transition metals may result in the development of stimuli-responsive devices, highlighting the role of cooperative metal–photoswitch interactions in tailoring specific material properties and design applications [77].

### 5.3. Postsynthetic Chemistry of Complex Structures

One of the most challenging trends in supramolecular chemistry is the synthesis of topologically complex structures, such as bonds and knots. In this select group are Solomon bonds based on the assembly of cyclic helicates by metal quenching (Cu^+^ or Li^+^), followed by covalent cyclization reactions [78]. Likewise, in 1999 Fujita and Sauvage [79] reported the quantitative formation of a Solomon bond by self-assembly. Schouwey et al. [80] synthesized ligands with 2,2′-bipyridine and Cu^+^ cores and pyridine groups for Pt^2+^ coordination. In detail, the authors describe the quantitative synthesis of a molecular Solomon bond from 30 subcomponents. The resulting structure is formed by the assembly of 12 cis-blocked Pt^2+^ complexes, 6 Cu^+^ ions, and 12 rigid N-donor ligands. This favors the interlocking of two rings and six repeating Pt(ligand) units, while the six Cu^+^ ions connect the two rings.

A pioneering work on the construction of the heteroleptic Solomon bond framed by two distinct metalamacrocycles is linked to the process of topological transformation based on a supramolecular fusion involving three distinct topological structures [81].

The research team led by Jin [82] presented an overview of molecular knots comprising a single closed strand and their substantial differences in terms of molecular bonds consisting of multiple rings mechanically intertwined with each other. They point out that the topologies are well described using the Alexander-Briggs notation [83], where the descriptor $X_{z}^{y}$ is used to denote a specific topology. *X* corresponds to the minimum number of crossings in the projection of the topology, *y* is the number of constituent rings, and *z* is the order of the given topology among its pairs with the same descriptors *X* and *y*.

For Jin and his team [84], the construction of non-trivial mechanically interlocked molecules (MIMs) has also been inherent and attractive. They have consequently outlined the molecular construction of closed three-link chains ($6_{1}^{3}$ links) (Figure 11A), for instance, the construction of [3]catenane with six molecular linkages. They detail that since synthesis from three rings is extremely difficult and the probability remains during the self-assembly process that alternative topological isomers, such as $6_{2}^{3}$ links (Figure 11B), cyclic [3]catenanes ($6_{3}^{3}$ links) (Figure 11C), and linear [3]catenanes ($4_{1}^{3}$ links) (Figure 11D) and other included topologies, will arise. Among the knotted molecules, unlike the faster-growing trefoil knots (Figure 11E), the performance of figure-eight knots ($4_{1}$ knots, single strand with four crossings) (Figure 11F) is inauspicious.

As already outlined through this review, coordination-driven self-assembly is a reliable strategy in the elaboration of supramolecular architectural structures ascribable to high mechanical strength, and to an elasticity and adaptability of metal–ligand coordination bonds.

The greatest improvement in the strategies for transforming molecular topologies via Diels–Alder click reactions was demonstrated by Tang et al. [85]. This feature details the reaction of a bis(3-(pyridin-4-yl)phenyl)-1,2,4,5-tetrazine ligand synthesized by the Suzuki coupling method with reactive tetrazine moieties. This ligand, in turn, reacts with the organometallic units [Cp*_2_M_2_(*μ*-TPPHZ)(OTf)_2_](OTf)_2_ (M = Rh^III^, or Ir^III^) to yield close-packed molecular trefoil knots. To create the molecular Solomon bonds, the steric position of trefoil knots was modulated in three dimensions via an inverse electron-demand Diels–Alder (IEDDA) reaction. It should be emphasized that the IEDDA reaction does not hinder metal–ligand coordination and has compatibility with a miscellaneous of dienophiles.

## 6. Conclusions

The development of new technologies with particularly systematic approaches to forming structures with a variety of metals is a key issue within metallo-supramolecular chemistry. In this review, the conformational foundations of MOCs and their similarity to MOPs are given. As the intrinsic porosity units for the construction of networks with perfectly structured pores, MOCs are often alternatively referred to as MOPs.

The molecular geometry of supramolecular systems is as important as how the MOCs are structured, and there are readily accessible sources of information corresponding to studies of metal nodes and organic connectors or linkers. Unlike other indices, connector-based geometry with π-electron systems, such as MOC molecules, can be a link that favors families of symmetric or non-symmetric compounds.

For a long time, it was the expectation and goal of chemists to achieve a characterization of ring (aromatic) compounds in a simple way, namely with a single scale. This view was challenged by the development of two distinct types of facial building blocks, characterized by clockwise/anticlockwise and plus/minus rotational patterns. The assembly of these facial units into FRP has resulted in the creation of numerous chiral cages with remarkable structural tunability and stereoselectivity [9]. In addition, several of the studies described point towards complex systems that are also examined. As we look to the future, the possible discoveries and applications that await us in the very dynamic fields of research in self-assembly, supramolecularity, and their MOC relationship are, in themselves, encouraging, and, of course, will be achieved through innovation and interdisciplinary approaches.

## Figures and Tables

**Figure 1 molecules-30-00462-f001:**
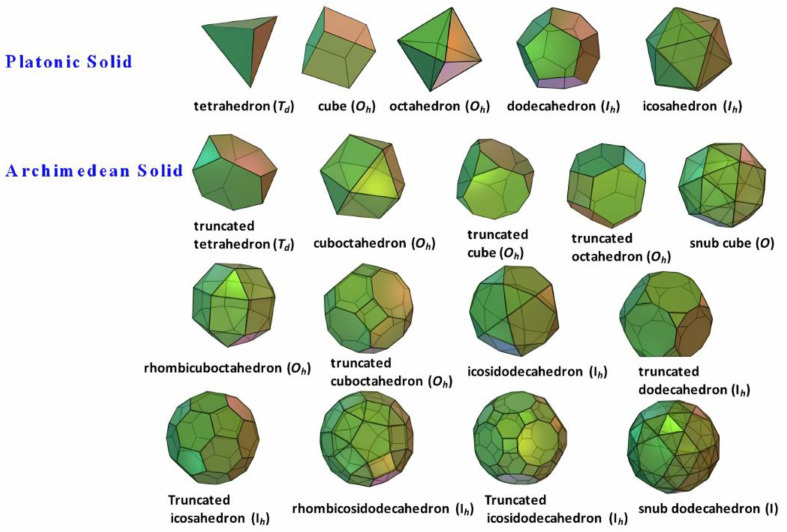
Illustration of five Platonic solids and thirteen Archimedean solids, their corresponding symmetry groups are in parentheses. Reproduced with permission from Pan et al. [8]. Copyright 2017 Elsevier B.V.

**Figure 2 molecules-30-00462-f002:**
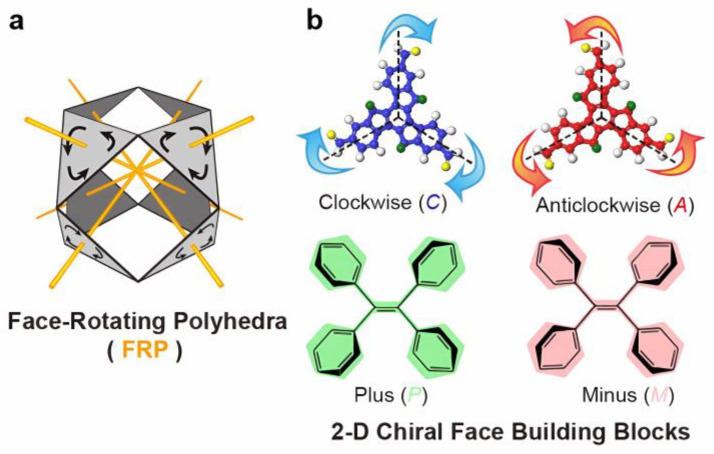
(**a**) The face-rotating polyhedra (FRP) concept proposed by Buckminster Fuller. (**b**) Two types of building blocks illustrating clockwise (C) or anticlockwise (A) patterns, as well as plus (P) or minus (M) patterns resulting from nonplanar conformations. Reproduced with permission from Dong et al. [9]. Copyright 2024 American Chemical Society.

**Figure 3 molecules-30-00462-f003:**
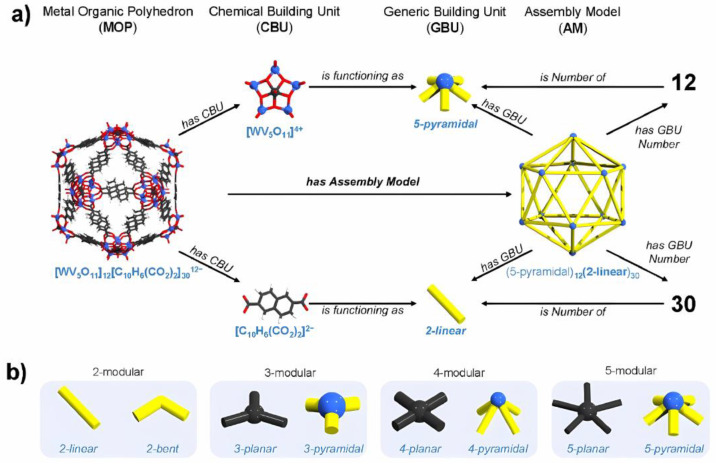
(**a**) Relations between MOP, CBUs, GBUs, and assembly models. (**b**) Four general types of GBUs. Reproduced with permission from Kondinski et al. [19]. Copyright 2022 American Chemical Society.

**Figure 4 molecules-30-00462-f004:**
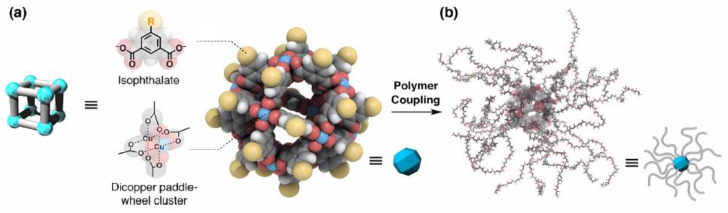
(**a**) Structure of a typical isophthalate-based MOP with a functional group at the fifth position of the isophthalate ligand. Cu, blue; C, gray; O, red; H, white; functional group R, yellow. (**b**) Polymer-conjugated MOP as a cavity module for the bottom–up design of porous soft materials. Reproduced with permission from Hosono et al. [27]. Copyright 2018 American Chemical Society.

**Figure 5 molecules-30-00462-f005:**
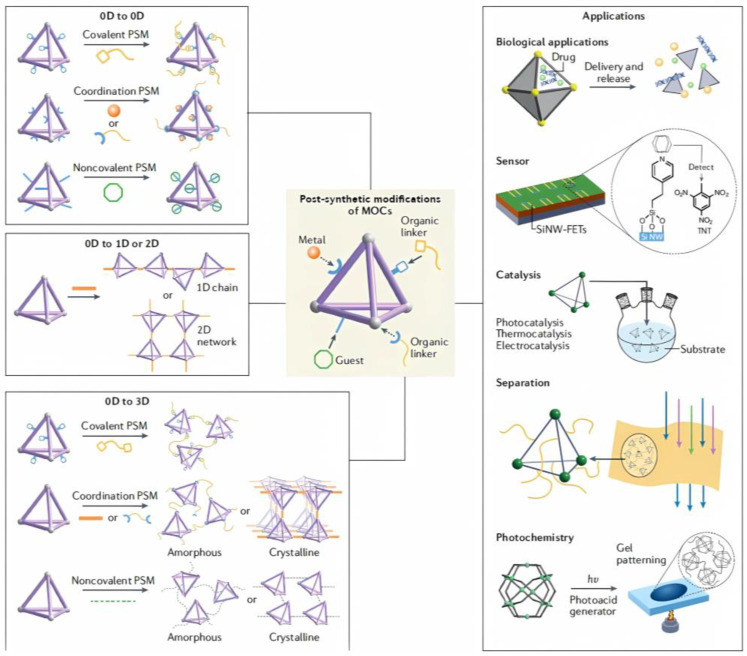
Strategies for post-synthetic modifications of metal–organic cages and the targeted properties. Strategies encompass various dimensional changes and/or functional group modifications that can expand the scope of applications. MOCs, metal–organic cages; PSM, post-synthetic modification; SiNW-FETs, silicon nanowire-based field-effect transistors. Reproduced with permission from Liu et al. [44]. Copyright 2022 Springer Nature Limited.

**Figure 6 molecules-30-00462-f006:**
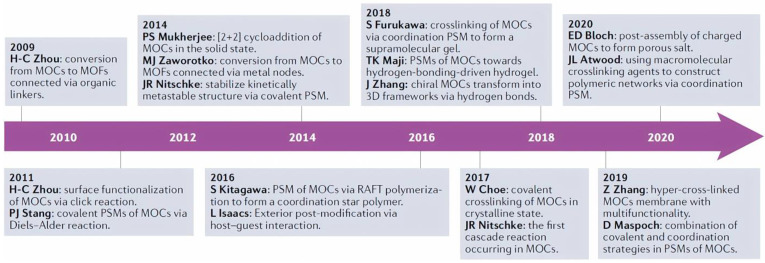
The chronological development in post-synthetic modifications of metal–organic cages with representative examples. MOCs, metal–organic cages; MOFs, metal–organic frameworks; PSM, post-synthetic modifications; RAFT, reversible addition–fragmentation chain transfer. Reproduced with permission from Liu et al. [44]. Copyright 2022 Springer Nature Limited.

**Figure 7 molecules-30-00462-f007:**
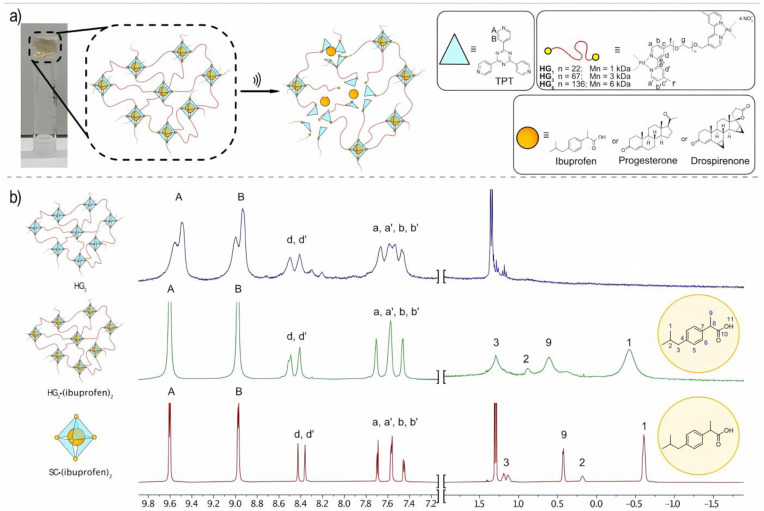
(**a**) Formation of metal–organic cage-crosslinked polymer hydrogels by terminal functionalization of PEG with bipyridine ligands, with hydrogels formed from the corresponding palladium nitrate precursors in combination with 2,4,6-tri(4-pyridyl)-1,3,5-triazine (TPT); (**b**) ^1^H NMR (600 MHz, D_2_O) of hydrogels HG_3_ without a guest (top), hydrogel HG_3_ (ibuprofen)_2_ encapsulating two ibuprofen moieties (middle) and SC· (ibuprofen)_2_. Reproduced with permission from Küng et al. [68]. Copyright 2023 Wiley-VCH GmbH.

**Figure 8 molecules-30-00462-f008:**
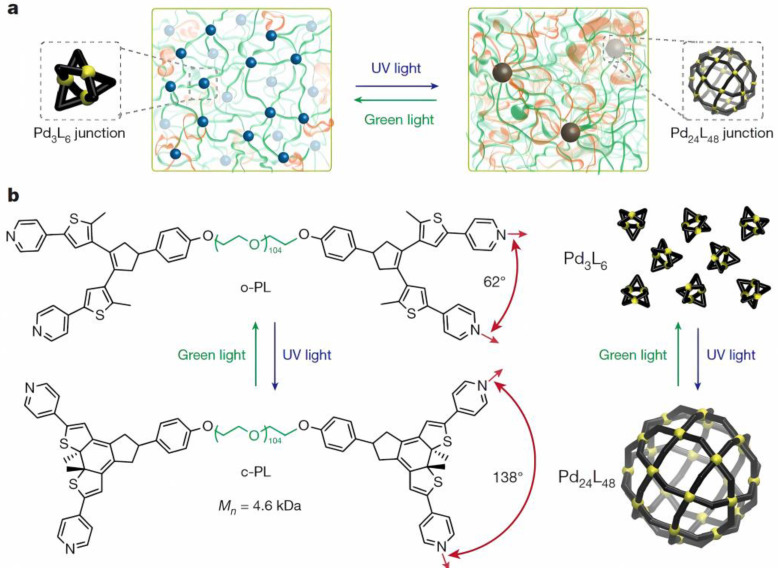
Design of polyMOCs with photoswitchable topology. (**a**) Schematic illustration of photo-regulated interconversion between two different network topologies. Photoresponsive MOCs are introduced as junctions within polyMOCs. Upon UV irradiation, the MOC rearranges its structure from Pd_3_L_6_ to Pd_24_L_48_. Reversal of the MOC structure with green light regenerates the original network topology. (**b**) Chemical structure of the photoresponsive polymer ligand and a schematic of MOC interconversion. Dithienylethene (DTE) moiety undergoes electrocyclic ring-closure and ring-opening upon UV and greenlight irradiation, respectively. Photoinduced ring closure/opening leads to a change in the bite angle between the two attached pyridine groups. Hence, the open-form (o-PL) and closed-form (c-PL) polymer ligands form small Pd_3_(o-PL)_6_ and large Pd_24_(c-PL)_48_ MOCs, respectively, in the presence of Pd^2+^. Reproduced with permission from Gu et al. [72]. Copyright 2018 Springer Nature Limited.

**Figure 9 molecules-30-00462-f009:**
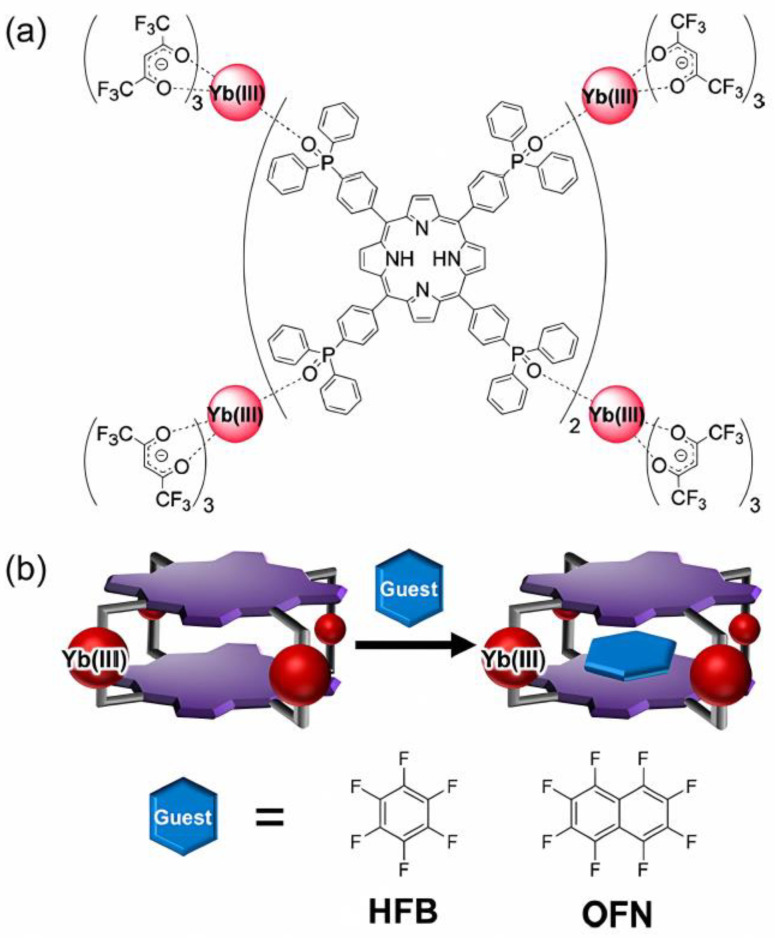
(**a**) Molecular structure of [Yb_4_(hfa)_12_(PorTPPO)_2_]. (**b**) Schematic illustration of host–guest complexation. Reproduced with permission from Hosoya et al. [75]. Copyright 2024 American Chemical Society.

**Figure 10 molecules-30-00462-f010:**
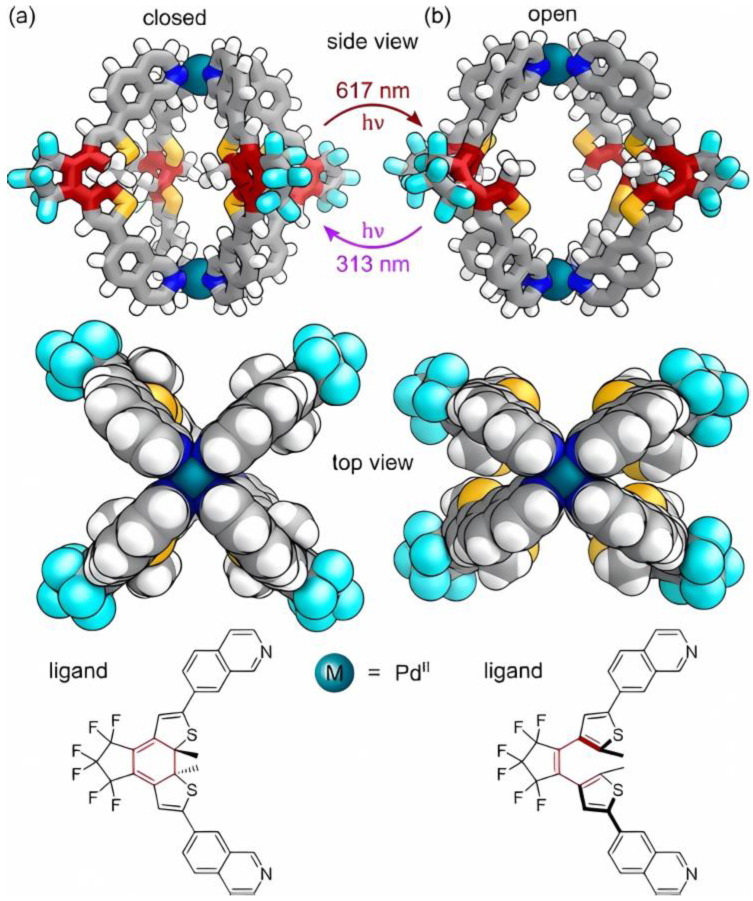
Differentiation between isomeric MOCs: (**a**) closed MOC M_2_L_4_ and (**b**) open MOC M_2_L_4_ are interconvertible isomeric structures by UV/visible light. Reproduced with permission from Pilgrim and Champness [76]. Copyright 2020 Wiley-VCH GmbH.

**Figure 11 molecules-30-00462-f011:**
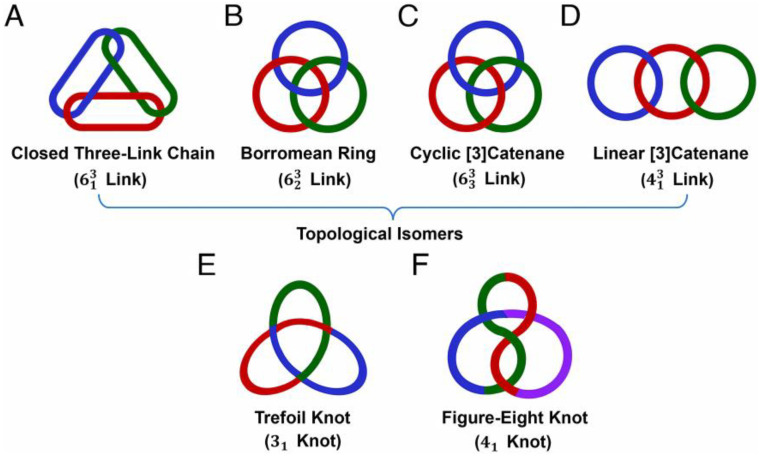
Topological structures of molecular links and knots. (**A**) Closed three-link chain ($6_{1}^{3}$ link). (**B**) Borromean ring ($6_{2}^{3}$ link). (**C**) Cyclic [3]catenane ($6_{3}^{3}$ link). (**D**) Linear [3]catenane ($4_{1}^{3}$ link). (**E**) Trefoil knot ($3_{1}\;$knot). (**F**) Figure-eight knot ($4_{1}$ knot). Reproduced with permission from Bao et al. [84]. Copyright 2024 PNAS.

## Data Availability

No new data were created or analyzed in this study. Data sharing is not applicable.

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
