# Peer review of "Functional Post-Synthetic Chemistry of Metal–Organic Cages According to Molecular Architecture and Specific Geometry of Origin"

_molecules, 2025, doi:10.3390/molecules30030462_

Round 1
Reviewer 1 Report
Comments and Suggestions for Authors
This manuscript introduces the structural characteristics of MOCs and the functional modification after synthesis of MOCs. The content of the paper is basically complete, but the discussion is not deep enough. The author needs to enrich the content and sort out the context of the paper, and the focus is more obvious. after some explanation and modification, I recommend this job.
1. Add subheadings to make it more organized.
2. Please use more illustrations to understand the content of the article more clearly and intuitively.
3. Please add more examples for further discussion.
Reviewer 2 Report
Comments and Suggestions for Authors
In the submitted review, the authors present various strategies for the design, synthesis, and post-synthetic modification of metal-organic cages. The manuscript begins with an in-depth discussion of the structural features of these systems, followed by an overview of recent advancements in (post-)synthetic approaches. Additionally, their specificity and application in different fields is explored. Based on the cited references, I believe that the manuscript has the potential to contribute meaningfully to the field. However, I am unable to recommend it for publication in its current form for the following reasons:
- The special issue focuses on the synthesis and application of metal-organic frameworks (MOFs), yet the manuscript primarily discusses metal-organic cages (MOCs), which are discrete molecular entities in contrast to the extended structures of MOFs. The manuscript would be more aligned with the issue’s theme if it included a more comprehensive discussion comparing PSM strategies for MOFs and MOCs.
- The manuscript is challenging to read due to its heavy tone and lack of clear logical flow between sections. In several parts, the cited references appear to be disconnected, which detracts from the coherence of the argument. A more structured approach would help guide the reader through the content more effectively.
- The manuscript requires careful revision of the English language to ensure clarity, fluency, and precision.
Additionally, I suggest considering the inclusion of the following recent references:
- Park et al., Small 2024, 20, 2308393, https://doi.org/10.1002/smll.202308393
- Gan et al., Angew. Chem. Int. Ed. 2024, 63, e202410731, https://doi.org/10.1002/anie.202410731
- Liu et al., Chem. Eur. J. 2024, 30, e202402499, https://doi.org/10.1002/chem.202402499
- Zhou et al., Chem. Eur. J. 2024, 30, e202400020, https://doi.org/10.1002/chem.202400020
Before resubmitting, I recommend a more thorough evaluation of the review’s structure and organization. To better highlight the various synthetic and post-synthetic strategies, I suggest incorporating further subsections. At present, the section on post-synthetic modifications (PSMs) is relatively limited, even though it should be a central focus of the manuscript. If the authors decide to maintain the focus on MOCs' synthesis and post-functionalization, I recommend condensing the introductory part on structural aspects, limiting it to the most significant works, and expanding the section on PSMs. Alternatively, if the authors wish to maintain a broader discussion, it would be better shifting the focus toward a more general examination of the design, structure, and synthetic strategies of MOCs.
Comments on the Quality of English LanguageEnglish language requires a careful revision for grammar and form (too long sentences, wrong use of puntuaction, ect.).
Reviewer 3 Report
Comments and Suggestions for Authors
The review by considers the peculiarities of metal-organic cages (or metal-organic polyhedral according to O. M. Yaghi) as prospective class of hybrid materials and methods of the post-synthesis modification of their metal nodes, linkers or host-guest chemistry) for better functional properties, i.e. catalytic or photoswitchable ones) A special emphasis has been done on the specificity of MOCs and differences between MOCs and MOFs (metal-organic frameworks as rather modern class of hybrid nanoporous solids).
The reference list includes 68 publications, most of them are belonging to the period of 2021-2024 years. This manuscript will be interesting for a broad readership, and may be published after minor revision by addressing the following issues.
1. There are some technical errors in the paper. For instance, P. 3. L. 113, 114. “An alternative to the permanent porosity of polyhedral organometallic structures has been published by Yagui and his team of collaborators.” The correct variant is metal-organic and Yaghi.
2. The quality of Figures in manuscript should be improved.
‘
Reviewer 4 Report
Comments and Suggestions for Authors
The review manuscript by Rodrigo Cué-Sampedro and José Antonio Sánchez-Fernández describes advances in the field of postsynthetic chemistry of metal-organic cages. The manuscript is interesting to the molecular cage community; however, it needs better organization and general improvements. After a detailed review, the manuscript might be of interest to Molecules readers after introducing the following important corrections.
The review does not include any original figures, as all figures are adapted from the literature. While this is not a negative aspect, it highlights that there has been somewhat insufficient effort in preparing original graphic material for the review. The authors should improve the review by adding original figures that illustrate the key concepts described in this review, highlighting what distinguishes it from other similar reviews in the literature.
The review should include the structures of all cages described in the examples. In this regard, some examples are only described in the text and not depicted in any figures, making the review difficult to follow. Furthermore, the included figures do not present the structures of molecular cages, and these cage structures should be added. The structures of all cages in the manuscript should be numbered in both the text and the figures, allowing readers to follow the text while reviewing the cage figures.
Figures 3, 4, 5, 6: are too small, please increase the size to read all the details (now some details are not possible to see)
A figure and a narrative illustrating the concept of postsynthetic chemistry of metal-organic cages are more or less adequately explained in the text of Section 4 and Figure 4. However, Figure 4 needs to be improved, or an additional figure should be included, incorporating examples of postsynthetic modification.
The text on the lines 95-140 and the Figure 2 are not clearly related to postsynthetic modification. Authors should consider whether this information is necessary for the review or if it should be removed.
Both the title of the review, the abstract, and the introduction should clearly state the aim of the review. For example, reading the title and introduction gives the impression that the review will detail synthetic procedures to achieve post-synthetic modification of metal-organic cages. However, the review provides very little detail on how these post-synthetic modifications are performed, with limited detail on the synthetic chemistry.
The manuscript needs to be revised to correct grammatical errors, and several typographical errors found throughout the text.
Reviewer 5 Report
Comments and Suggestions for Authors
Authors:
I have reviewed your manuscript on functional metal-organic cages, which is understood to be a review paper.
Abstract and keywords: Concise and informative, fully understandable.
Introduction: It gives sufficient details on the state-of-the-art in the given field. Enough references given, i.e., this part of the manuscript is well written and it gives all relevant items of information which should be presented to introduce the review paper.
Structural criteria of metal-organic polyhedrons: This part of the review is also well written. Adequate figures graphically explain the basis of the investigations done in this innovative field.
Criteria for the synthesis of metal-organic cages and their structural post-modifications: These two paragraphs are very important to understand what was done in this field, and which are the most important findings. This part is again accompanied by the instructive figures.
Specificity of metal-organic cages: This part of the review deals with the ability of different ligands and metal ions in a formation of different types of complex self-linked molecular networks. Different architecture of the space objects is presented and discussed.
Conclusion: It summarized the most important items of information given in this review paper.
References: Their number id adequate to the review paper, and they are cited in an agreement with the journal's requirements.
Copyright for the reproduced figures: The authors presented the copyright permissions to reproduce important figures from the original articles.
Suggestion: Accept this manuscript in its current version for publication in the journal Molecules.
Round 2
Reviewer 1 Report
Comments and Suggestions for Authors
In the revised manuscript entitled “Functional post-synthetic chemistry of metal-organic cages according to its molecular architecture and specific geometry of origin’ comments have been seriously treated and properly responded. I therefore recommend the acceptance of this paper for publication in Molecules essentially as is.
Author Response
Dear Reviewer,
We appreciate your comments that encourage us to continue with our projects. We appreciate how rewarding they are.
Reviewer 2 Report
Comments and Suggestions for Authors
In the revised version of the manuscript, additional subsections were included, and some parts of the text were rewritten. The suggested references have also been incorporated. While these revisions have led to some improvement, I am still unable to recommend the manuscript for publication in Molecules journal.
The issues related to the overly heavy form and the English language listed in the previous report have not been sufficiently addressed. A comprehensive revision of the manuscript is required before it can be considered for resubmission.
Comments on the Quality of English LanguageEnglish language requires a careful revision for grammar and form (too long sentences, wrong use of puntuaction, ect.
Reviewer 4 Report
Comments and Suggestions for Authors
The review manuscript by Rodrigo Cué-Sampedro and José Antonio Sánchez-Fernández describes advances in the field of postsynthetic chemistry of metal-organic cages. The manuscript has minimally improved, with some of the requested changes not implemented. The manuscript might be of interest to Molecules readers after introducing the following important corrections requested in the previous round of review and not implemented.
The review should include the structures of all cages described in the examples. In this regard, some examples are only described in the text and not depicted in any figures, making the review difficult to follow. Furthermore, the included figures do not present the structures of molecular cages, and these cage structures should be added. The structures of all cages in the manuscript should be numbered in both the text and the figures, allowing readers to follow the text while reviewing the cage figures.
Answer 2. The text has a logical and structured sequence, and the figures are in line with the full text. As this is a review, we do not consider it feasible to number the structures: if the text is comprehensible, because if it is, the figures are only a support.
>> The answer does not justify the importance of adding all the structures of the cages, they must be added into the figures
Figures 3, 4, 5, 6: are too small, please increase the size to read all the details (now some details are not possible to see).
Answer 3. The resolution of the Figures is as required by Molecules journal. The Figures certainly retain the original details.
>> The text of the figures is too small, at least the size of the image must be increased.
